# Targeting the Pentose Phosphate Pathway for SARS-CoV-2 Therapy

**DOI:** 10.3390/metabo11100699

**Published:** 2021-10-13

**Authors:** Denisa Bojkova, Rui Costa, Philipp Reus, Marco Bechtel, Mark-Christian Jaboreck, Ruth Olmer, Ulrich Martin, Sandra Ciesek, Martin Michaelis, Jindrich Cinatl

**Affiliations:** 1Institute for Medical Virology, University Hospital, Goethe University, 60596 Frankfurt am Main, Germany; Denisa.Bojkova@kgu.de (D.B.); Philipp.Reus@itmp.fraunhofer.de (P.R.); Marco.Bechtel@kgu.de (M.B.); Sandra.ciesek@kgu.de (S.C.); 2Copenhagen Hepatitis C Program (CO-HEP), Department of Infectious Diseases, Hvidovre Hospital and Department of Immunology and Microbiology, University of Copenhagen, 1455 Copenhagen, Denmark; rcosta@sund.ku.dk; 3Fraunhofer Institute for Translational Medicine and Pharmacology (ITMP), Theodor-Stern-Kai 7, 60590 Frankfurt am Main, Germany; 4Leibniz Research Laboratories for Biotechnology and Artificial Organs (LEBAO), Hannover Medical School, Carl-Neuberg-Str. 1, 30625 Hannover, Germany; Jaboreck.Mark-Christian@mh-hannover.de (M.-C.J.); Olmer.Ruth@mh-hannover.de (R.O.); Martin.Ulrich@mh-hannover.de (U.M.); 5Member of the German Lung Research Center (DZL), Feulgenstrasse 12, 35392 Giessen, Germany; 6German Center for Infection Research, DZIF, External Partner Site, 60596 Frankfurt am Main, Germany; 7School of Biosciences, University of Kent, Canterbury CT2 7NJ, UK

**Keywords:** SARS-CoV-2, COVID-19, antiviral therapy, pentose phosphate pathway, oxythiamine, benfooxythiamine, 2-deoxy-d-glucose

## Abstract

SARS-CoV-2 is causing the coronavirus disease 2019 (COVID-19) pandemic, for which effective pharmacological therapies are needed. SARS-CoV-2 induces a shift of the host cell metabolism towards glycolysis, and the glycolysis inhibitor 2-deoxy-d-glucose (2DG), which interferes with SARS-CoV-2 infection, is under development for the treatment of COVID-19 patients. The glycolytic pathway generates intermediates that supply the non-oxidative branch of the pentose phosphate pathway (PPP). In this study, the analysis of proteomics data indicated increased transketolase (TKT) levels in SARS-CoV-2-infected cells, suggesting that a role is played by the non-oxidative PPP. In agreement, the TKT inhibitor benfooxythiamine (BOT) inhibited SARS-CoV-2 replication and increased the anti-SARS-CoV-2 activity of 2DG. In conclusion, SARS-CoV-2 infection is associated with changes in the regulation of the PPP. The TKT inhibitor BOT inhibited SARS-CoV-2 replication and increased the activity of the glycolysis inhibitor 2DG. Notably, metabolic drugs like BOT and 2DG may also interfere with COVID-19-associated immunopathology by modifying the metabolism of immune cells in addition to inhibiting SARS-CoV-2 replication. Hence, they may improve COVID-19 therapy outcomes by exerting antiviral and immunomodulatory effects.

## 1. Introduction

Severe acute respiratory syndrome coronavirus 2 (SARS-CoV-2) is causing the ongoing coronavirus disease 2019 (COVID-19) pandemic [1,2], which has so far resulted in more than 200 million confirmed COVID-19 cases and more than 4.3 million confirmed COVID-19-associated deaths [3].

Metabolic disorders and diabetes are associated with an increased risk of severe COVID-19 [4,5,6]. Increased amounts of glucose in the sera of COVID-19 patients were correlated with poor prognosis in individuals both with and without pre-existing diabetes [7]. At the cellular level, SARS-CoV-2 induces a shift of the host cell metabolism towards glycolysis in infected cells. The glycolysis inhibitor 2-deoxy-d-glucose (2DG), which targets hexokinase (the rate-limiting enzyme in glycolysis), interferes with SARS-CoV-2 infection in colon adenocarcinoma (Caco-2) cells and monocytes [8,9], suggesting that the observed metabolic changes support virus replication and represent an antiviral drug target.

Based on these findings, a clinical trial (CTRI/2021/01/030231) investigating 2DG in COVID-19 patients has begun in India [10,11]. Moreover, the Swiss company DG-Nika AG has developed a pocket inhaler for the treatment of COVID-19 patients with 2DG [12], and the U.S. company Moleculin has developed the 2DG derivative WP1122 as a COVID-19 treatment [13].

Changes in glycolysis can also affect other metabolic pathways. Metabolic intermediates generated by the glycolytic enzymes supply both the oxidative and the non-oxidative branches of the pentose phosphate pathway (PPP) [14]. The non-oxidative PPP branch converts glycolytic intermediates into ribose-5-phosphate, required for the synthesis of nucleic acids, as well as sugar phosphate precursors that are necessary for the synthesis of amino acids [14]. Moreover, SARS-CoV-2 infection was associated with changes in the PPP in a ferret model [15]. Hence, in this study, we investigated whether the anti-SARS-CoV-2 activity of 2DG can be increased by additionally targeting the non-oxidative PPP.

## 2. Results

### 2.1. SARS-CoV-2 Infection Affects Key Enzymes of the Glycolysis Pathway and Non-Oxidative Pentose Phosphate Pathway (PPP)

Initially, we characterized metabolic changes associated with SARS-CoV-2 infection using a proteomics dataset derived from SARS-CoV-2-infected Caco-2 cells [8] (Figure 1A). Cellular levels of key drivers of glycolysis including hexokinase 1 and 2 (HK1 and HK2) were increased in SARS-CoV-2-infected cells, while negative regulators such as fructose-1,6-bisphosphatase 1 and 2 (FBP1 and FBP2) were reduced (Figure 1B,C).

### 2.2. Increased Glycolytic Activity in SARS-CoV-2-Infected Cells

In agreement with the findings from the analysis of proteomics data, glycolytic activity was elevated in SARS-CoV-2-infected cells (Figure 2). SARS-CoV-2-infected cells displayed a slight, non-significant reduction in mitochondrial ATP production and a significant increase in glycolytic ATP production (Figure 2A). In concert with these findings, virus infection resulted in minor effects on oxidative phosphorylation (Figure 2B) but significantly increased glycolytic activity (Figure 2C). These findings were confirmed using a Seahorse XF Mito Stress Test Kit (Figure 2D,E). Altogether, SARS-CoV-2 infection resulted in a metabolic shift towards enhanced glycolytic activity (Figure 2E).

### 2.3. Inhibition of the Non-Oxidative Pentose Phosphate Pathway (PPP) Interferes with SARS-CoV-2 Replication

Changes in glycolysis can impact the non-oxidative PPP [14]. Moreover, transketolase (TKT) and transaldolase 1 (TALDO1), two constituents of the non-oxidative PPP branch [14], displayed increased levels in SARS-CoV-2-infected cells (Figure 1B,C), suggesting that SARS-CoV-2 infection also affects the non-oxidative PPP branch. As such, we next investigated whether a TKT inhibitor may affect SARS-CoV-2 replication.

Oxythiamine is an inactive analog of thymine that irreversibly inhibits TKT [16]. Here, we used benfooxythiamine (BOT), an oxythiamine prodrug [17], for our experiments. Indeed, non-toxic BOT concentrations reduced cellular SARS-CoV-2 Spike (S) protein levels in Caco2 cells infected with two different SARS-CoV-2 isolates (FFM1 and FFM7) in a concentration-dependent manner (Figure 3A–C). Moreover, BOT inhibited replication of SARS-CoV-2 FFM7 in air–liquid interface (ALI) cultures of primary bronchial epithelial cells as indicated by the quantification of viral genomic RNA copy numbers (Figure 3D–F).

### 2.4. Benfooxythiamine (BOT) Increases the Anti-SARS-CoV-2 Effects of 2-Deoxy-d-Glucose (2DG)

Here, 2DG and BOT both interfere with the non-oxidative PPP [14]. In addition, 2DG inhibits the glycolysis enzymes hexokinase and phosphoglucose isomerase (PGI) that are required for the synthesis of fructose-6-phosphate, which serves as a glycolytic intermediate for generating ribose-5-phosphate in the non-oxidative PPP branch [14]. BOT inhibits TKT, a crucial player in the non-oxidative PPP [14,16,17]. Thus, the combination of both compounds may result in increased anti-SARS-CoV-2 activity.

Indeed, the antiviral effects of BOT were further increased by the glycolysis inhibitor 2DG, as indicated by cellular S protein levels (Figure 4A,B) and viral genomic RNA copy numbers (Figure 4C).

## 3. Discussion

In this study, we show that the TKT inhibitor BOT inhibits SARS-CoV-2 replication and increases the anti-SARS-CoV-2 activity of the glycolysis inhibitor 2DG. 2DG and its derivative WP1122 are under development for use in the treatment of COVID-19 patients [10,11,12].

TKT is a crucial player in the non-oxidative PPP [14]. Proteomics data indicated increased TKT levels in SARS-CoV-2-infected cells, suggesting a role is played by the non-oxidative PPP. Notably, activity of the non-oxidative PPP may not always be a direct consequence of available TKT levels, as TKT activity can be strongly increased by heterodimer formation with transketolase-like 1 (TKTL1) [18]. However, anti-SARS-CoV-2 effects exerted by the TKT inhibitor BOT suggest that TKT and the non-oxidative PPP are functionally involved in SARS-CoV-2 replication.

Additionally, TKT activity is enhanced in diabetes patients [19,20,21] and diabetes patients are at a high risk from COVID-19 [4,5,6]. Given the potential role of TKT in SARS-CoV-2 replication identified here, elevated TKT levels may be one of the factors that predispose diabetes patients to experience severe COVID-19. Moreover, over-activation of the PPP has been suggested to drive hyperglycemia-associated vascular damage [22], and vascular damage is an integral part of COVID-19 pathology [23]. Thus, hyperglycemia-related PPP activity may further predispose diabetic patients to experience severe COVID-19.

Mechanistically, BOT and 2DG both reduce the cellular levels of ribose-5-phosphate. BOT directly inhibits TKT in the non-oxidative PPP (Figure 4C) [14,16,17]. 2DG inhibits the glycolysis enzymes hexokinase and phosphoglucose isomerase (PGI), both required for synthesis of fructose-6-phosphate, which serves as a glycolytic intermediate for generating ribose-5-phosphate in the non-oxidative PPP branch (Figure 4C) [14]. Furthermore, 2DG decreases cellular levels of glucose-6-phosphate, which serves as a precursor of ribose-5-phosphate in the oxidative PPP branch [14]; the oxidative PPP branch also seems to be activated in SARS-CoV-2-infected cells, as indicated by increased 6-phoshogluconate dehydrogenase (PGD) levels (Figure 1B,C). Notably, uncontrolled glucose and pentose phosphate levels correlate with poor prognosis and higher mortality in COVID-19 patients [24,25,26]. Hence, BOT could increase the effects of 2DG by further reducing ribose-5-phosphate levels (Figure 4C). Moreover, the glycolysis pathway and the non-oxidative PPP are further interlinked; for example, TKT depletion resulted in decreased levels of glycolytic enzymes including HK, PFK, and PKM2 in breast cancer cells [27].

In addition to providing additional evidence on how host cell metabolism is reprogrammed upon SARS-CoV-2 infection, our current findings are also of potential therapeutic relevance. Antiviral therapy options for SARS-CoV-2 infection/COVID-19 remain of high priority [2,28]. Vaccines dramatically reduce the proportion of severe COVID-19 cases [2,29]. However, it will take years until a substantial fraction of the world population will be vaccinated, and vaccine hesitation remains an issue [29]. Moreover, breakthrough infections occur in fully vaccinated individuals, and the formation of escape variants that are not well-covered by the immune response induced by current vaccines and/or previous infections represents an ongoing threat [30,31,32,33].

Immunomodulatory agents have had some success in reducing mortality in patients suffering from hyperinflammation [2,28,34,35,36]. Similarly, symptomatic treatments such as anticoagulants used for COVID-19-associated coagulopathy have improved therapy outcomes [2,28,37]. However, effective antiviral therapies are needed that prevent COVID-19 progression to advanced, life-threatening stages. In this context, the RNA polymerase inhibitor remdesivir and antibody preparations have been approved for the treatment of COVID-19 patients, but their efficacy appears to be limited [28]. Moreover, other direct antiviral approaches such as antisense strategies are under development for respiratory viruses such as SARS-CoV-2 [38,39,40] but are not yet clinically available as treatment options for COVID-19. Hence, effective additional antiviral treatment options are needed, and combinations of metabolic drugs such as 2DG and BOT may play a role in COVID-19 therapies in the future. In this context, strategies that target the host cell instead of viral targets have been postulated to be potentially associated with reduced resistance formation [41,42].

Notably, a metabolic shift towards glycolysis and PPP may also be involved in the inflammatory processes that are associated with COVID-19 progression to severe, life-threatening disease stages [9,43,44,45]. Moreover, reports show that a number of COVID-19 survivors suffer from metabolic disorders including abnormal glucose metabolism resulting in the metabolic derailment of immune cells [40]. Therefore, targeting the host cell metabolism by drug combinations such as BOT and 2DG may not only inhibit SARS-CoV-2 replication but also alleviate COVID-19 severity by immunomodulatory activity. Therefore, the further testing of such therapy strategies in animal models is warranted.

In conclusion, SARS-CoV-2 infection is associated with changes in the regulation of the PPP. The TKT inhibitor BOT inhibited SARS-CoV-2 replication, indicating a role of the non-oxidative PPP branch in the virus replication cycle. Moreover, BOT increased the activity of the glycolysis inhibitor 2DG, which is under development as a drug for the treatment of COVID-19 among other related glycolysis inhibitors [10,11,12]. Notably, metabolic drugs like BOT and 2DG may also interfere with COVID-19-associated immunopathology in addition to SARS-CoV-2 replication. Hence, they may improve COVID-19 therapy outcomes by exerting antiviral and immunomodulatory effects.

## 4. Materials and Methods

### 4.1. Proteomics Data

The LC–MS/MS proteomics data with the dataset identifier PXD017710 have been previously published [8]. Proteins derived from Caco-2 cells infected with SARS-CoV-2 after 24 h post infection and matching search terms “glycolysis” and “pentose phosphate pathway” in the KEGG database were selected for further analysis. Significantly deregulated proteins (*p* value ≤ 0.05) in mock versus infected cells were plotted using the heatmaps2 function of the gplots package of the R suite. Raw enrichment values were used to calculate per row Z scores. Maximum and minimum Z score were attributed the color red and blue respectively. Scripts are available upon request.

### 4.2. Measurement of ATP Rate

Caco-2 cells were seeded into a Seahorse XF96 V3 PS Cell Culture Microplate and grown until confluent. The cells were either mock-treated or infected with SARS-CoV-2/FFM7 for 24 h before measurement at an MOI of 0.01. The day before measurement, a sensor cartridge was hydrated with Seahorse XF calibrant solution at 37 °C in a non-CO_2_ incubator overnight. On the day of assay, the stock solutions of the Seahorse XF Real-Time ATP Rate Assay Kit (oligomycin, rotenone/antimycin A) were freshly prepared, diluted, and loaded into the corresponding sensor cartridge ports. Directly before measurement, the cells were washed with a warm Seahorse XF DMEM medium (pH 7.4, supplemented with 1 mM pyruvate, 2 mM glutamine, and 10 mM glucose) and subsequently incubated in this medium for 1 h at 37 °C in a non-CO_2_ incubator. The assay was conducted using the Seahorse XFe96 Analyzer and the Real-Time ATP Rate Assay Kit protocol. The order of injections was: oligomycin and rotenone/antimycin A. The final concentrations in the assay were 1.5 µM for oligomycin and 0.5 µM each for the rotenone/antimycin A-mix. The data was evaluated via Seahorse Wave Software and normalized to cell number via DAPI count of each well. Calculation of mito- and glyco-ATP was done via the Seahorse XF Real-Time ATP Rate Assay Report Generator.

### 4.3. Measurement of Mitochondrial Respiration and the Glycolytic Function

Caco-2 cells were infected with SARS-CoV-2/FFM7 or UV-treated SARS-CoV-2/FFM7 at an MOI of 0.01 for 24 h before measurement. On the day of assay, the stock solutions of the Seahorse XF Mito Stress Test Kit (oligomycin, FCCP, rotenone/antimycin A) and 2-DG were freshly prepared, diluted, and loaded into the corresponding sensor cartridge ports. Directly before measurement, the cells were washed with a warm Seahorse XF DMEM medium (pH 7.4, supplemented with 1 mM pyruvate, 2 mM glutamine and 10 mM glucose) and subsequently incubated in this medium for 1 h at 37 °C in a non-CO_2_ incubator. The assay was conducted using the Seahorse XFe96 Analyzer and a modified Mito Stress Test Kit protocol. The order of injections was oligomycin (1 µM), FCCP (20 mM), 2-DG (0.5 µM), and rotenone/antimycin A (0.5 µM). The data was evaluated via Seahorse Wave Software and normalized to cell number via DAPI count of each well.

### 4.4. Cell Culture and Virus Production

Caco-2 cells, colon carcinoma derived cell lines, were maintained in MEM (Minimal Essential Medium) containing 10% (*v*/*v*) foetal bovine serum, 10,000 U penicillin/streptomycin, and 2% (*v*/*v*) L-glutamine (Sigma Aldrich, Taufkirchen, Germany).

Primary human bronchial epithelial (HBE) cells were isolated from the lung explant tissue of a patient with lung emphysema as described previously [46]. The use of tissue was approved by the ethics committee of the Hannover Medical School (MHH, Hannover, Germany, number 2923–2015) and complied with the Code of Ethics of the World Medical Association. For differentiation into air–liquid interface (ALI) cultures, the cells were thawed and passaged once in PneumaCult-Ex Medium (StemCell technologies, Cologne, Germany) and then seeded on transwell inserts (12 well plate, Sarstedt, Nümbrecht, Germany) at 4 × 10^4^ cells/insert. After reaching confluence, medium on the apical side of the transwell was removed and medium in the basal chamber was replaced with PneumaCult ALI Maintenance Medium (StemCell Technologies) including Antibiotic/Antimycotic solution (Sigma Aldrich) and MycoZap Plus PR (Lonza, Cologne, Germany). Criteria for successful differentiation were the development of ciliary movement, an increase in transepithelial electric resistance, and mucus production.

Both SARS-CoV-2/FFM1 and SARS-CoV-2/FFM7 were isolated as previously described [47]. The viral stocks were produced by passaging virus on Caco-2 cells at MOI 0.1. Both strains underwent two passages. The virus titer was determined as TCID50/mL.

### 4.5. Antiviral and Cytotoxicity Assay

Caco-2 cells were seeded in a 96-well plate. After reaching confluency, cells were pre-treated with benfooxythiamine (BOT) (Zyagnum AG, Pfungstadt, Germany) for 24 h, and were additionally treated with 2-deoxy-d-glucose (2DG) or left untreated and infected with SARS-CoV-2 at MOI 0.01. Antiviral effects were determined by immunostaining (see Section 4.6) and/or quantification of viral genomes (see Section 4.7). Cytotoxic effects were determined by MTT assay as previously described [48].

HBE cells in ALI cultures were infected with SARS-CoV-2/FFM7 at an MOI of 1 from the apical site. The inoculum was removed after 2 h, and cells were washed three times with PBS. For antiviral drug testing, compounds were added after the infection period from both the apical and the basal site. The apical treatment was removed after one day. Genomic viral RNA copy numbers were determined after five days. Cytotoxicity was determined by LDH-Glo™Cytotoxicity Assay (Promega, Walldorf, Germany) according to the manufacturer’s protocol. Five days post infection, ALI cultures were washed apically with PBS for 30 min, and 10 µL of washing step solution was diluted 1/10 in a LDH storage buffer and stored at −80 °C until measurement. After thawing, measurement was performed using the plate reader Infinite 200 (Tecan).

### 4.6. Immunostaining

To detect the SARS-CoV-2 spike (S) protein in virus-infected cell cultures, cells were fixed with acetone:methanol (40:60) solution followed by incubation with a primary monoclonal antibody directed against SARS-CoV-2 S (1:1500, Sinobiological, via BIOZOL, Eching, Germany). Primary antibody binding was visualized using a peroxidase-conjugated anti-rabbit secondary antibody (1:1000, Dianova, Hamburg, Germany) and AEC substrate. Quantification of staining was performed using BIOREADER^®^-7000-F-z-I (Bio-Sys, Karben, Germany).

### 4.7. qRT-PCR of Viral Genome in Supernatants

RNA from cell culture supernatant was isolated using the QIAamp Viral RNA Kit (Qiagen, Hilden, Germany) according to the manufacturer’s instructions. The amount of viral RNA was detected by primers targeting the RNA-dependent RNA polymerase (RdRp): RdRP_SARSr-F2 (GTGARATGGTCATGTGTGGCGG) and dRP_SARSr-R1(CARATGTTAAASACACTATTAGCATA) using the Luna Universal One-Step RT-qPCR Kit (New England Biolabs) and a CFX96 Real-Time System C1000 Touch Thermal Cycler. The number of viral copies was determined using a standard curve generated by plasmid DNA (pEX-A128-RdRP) containing the corresponding amplicon regions of the RdRP target sequence.

### 4.8. Statistics

All experiments were performed in three independent replicates. GraphPad Prism 8 was used to prepare graphs and to perform statistical analyses. Statistical significance was calculated by a two-sided unpaired *t*-test.

## Figures and Tables

**Figure 1 metabolites-11-00699-f001:**
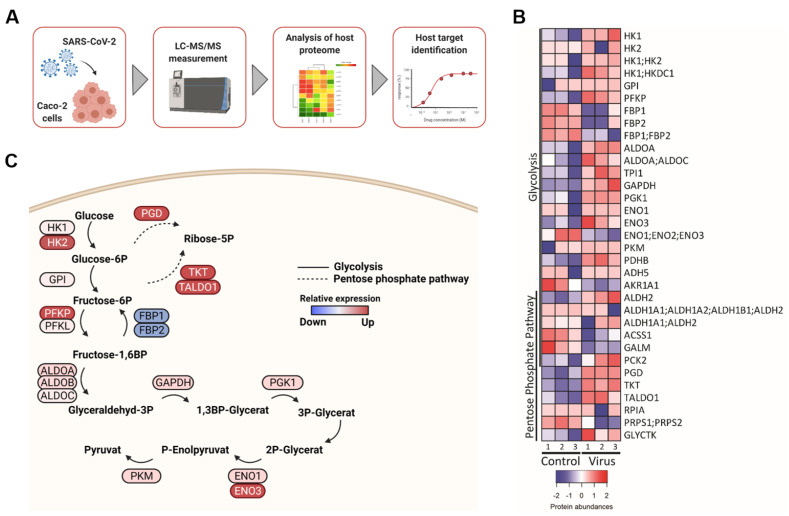
Glycolysis and pentose phosphate pathway (PPP) are deregulated during SARS-CoV-2 infection. (**A**) Schematic of proteomic analysis of SARS-CoV-2 infected Caco-2 cells. (**B**) Heatmap of changes in protein abundance of components of glycolysis and pentose phosphate pathway in SARS-CoV-2 infected Caco-2 cells at 24 h post infection. A Z score transformation was performed such that red and blue represent high and low protein abundance, respectively. The plot was performed using the heatmaps2 function of the gplots package of the R suite. (**C**) Pathway depiction of main regulators of glycolysis and PPP using BioRender.

**Figure 2 metabolites-11-00699-f002:**
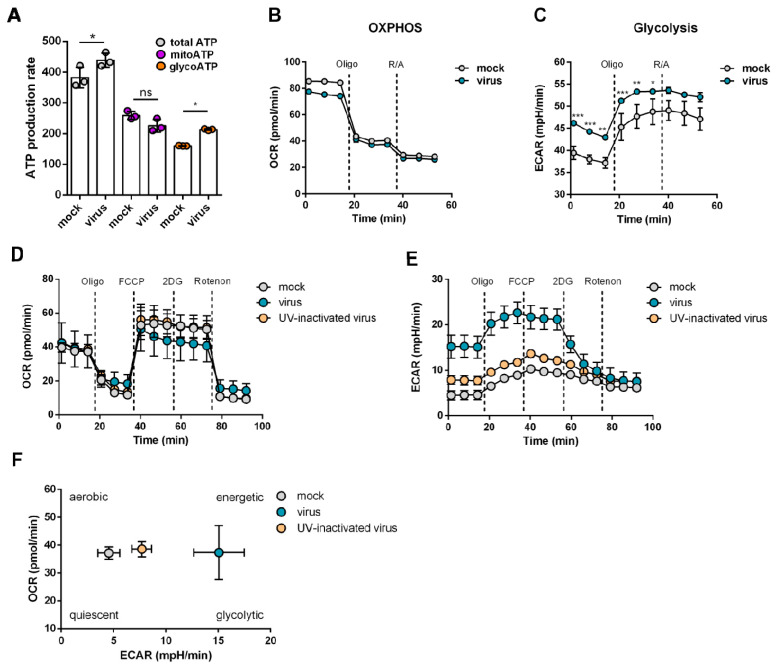
SARS-CoV-2 infection causes metabolic shift towards glycolysis. Caco-2 cells were infected with untreated or UV-inactivated SARS-CoV-2/FFM7 at an MOI of 0.01 for 24 h before measurement. (**A**–**C**) Pathway-specific ATP production (**A**) derived from mitochondrial respiration (**B**) and glycolysis (**C**) measurement. Mean + SD from three biological replicates are displayed. Statistical significance was determined through one way ANOVA. * *p* ≤ 0.05, ** *p* ≤ 0.01, *** *p* ≤ 0.001, ns: not significant. (**D**–**F**) Measurement of mitochondrial respiration and glycolytic function under stress conditions (Mito stress Test). Mean + SD from three biological replicates are displayed. ECAR, extracellular acidification rate; OCR, oxygen consumption rate.

**Figure 3 metabolites-11-00699-f003:**
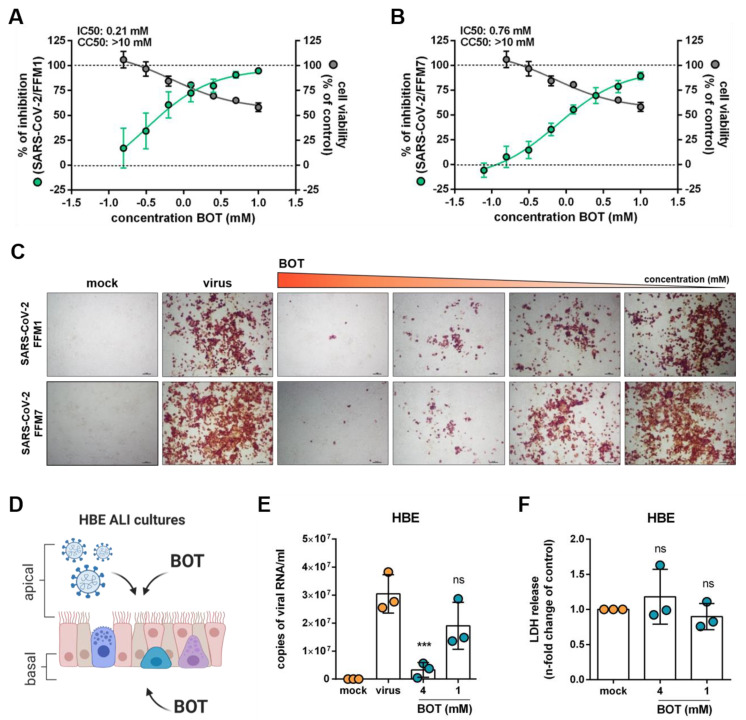
Targeting the pentose phosphate pathway (PPP) in SARS-CoV-2-infected cells. (**A**,**B**) Effects of the TKT inhibitor benfooxythiamine (BOT) on virus replication and cell viability in SARS-CoV-2 strain FFM1 and FFM7 (MOI 0.01)-infected Caco2 cells 24 h post infection. Caco-2 cells were pre-treated with different concentrations of BOT for 24 h. Percentage of viral inhibition was evaluated by spike (S) protein staining, and cell viability was measured by MTT assay. The IC50 and CC50 values were determined using the curve regression function of GraphPad Prism 8. Both plots represent mean ± SD of the three independent experiments. (**C**) Representative images illustrating SARS-CoV-2 S protein levels in SARS-CoV-2/FFM1 and SARS-CoV-2/FFM7 (MOI 0.01) infected Caco-2 cells treated with BOT 24h post infection. (**D**) BOT treatment strategy of air-liquid interface (ALI) cultures of primary human bronchial epithelial (HBE) cells infected with SARS-CoV-2/FFM7 (MOI 1). (**E**) Quantification of viral genomic RNA copy numbers in apical washes five days post infection. Bars represent mean ± SD of the three biological replicates. (**F**) Cytotoxicity assay measuring LDH levels in apical washes of BOT-treated HBE ALI cultures in comparison to untreated control. Bars display mean ± SD of the three biological replicates. *** *p* ≤ 0.001.

**Figure 4 metabolites-11-00699-f004:**
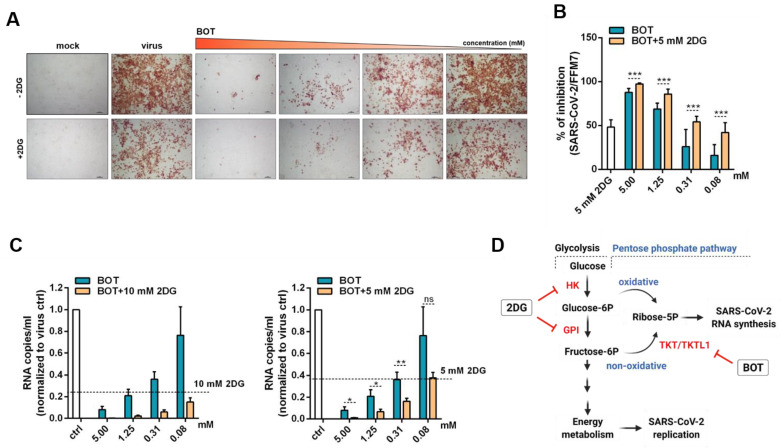
Inhibition of viral infection in benfooxythiamine (BOT)-treated cells in combination with 2-deoxy-d-glucose (2DG). Caco-2 cells were pre-treated with different concentration of BOT for 24 h. Then, the 2DG at concentration 5mM or 10 mM was added and cells were infected with SARS-CoV-2/FFM7 at MOI 0.01. (**A**) Representative images illustrating immunohistochemistry staining of SARS-CoV-2 spike protein in SARS-CoV-2/FFM7 infected Caco-2 cells treated with BOT in combination with 2DG. (**B**) Quantification of illustrating immunohistochemistry staining of SARS-CoV-2 spike protein in SARS-CoV-2/FFM7 infected Caco-2 cells treated with BOT in combination with 2DG. Values represent the mean ± SD of the three independent experiments. *p*-values were determined with a two-sided unpaired *t*-test. *** *p* ≤ 0.001 (**C**) Quantification of viral genomes in supernatant of SARS-CoV-2 infected Caco-2 cells treated with BOT in combination with 2DG or BOT alone. SARS-CoV-2/FFM7 RNA copy numbers used the RNA-polymerase (RdRp) gene by qRT-PCR of RdRp gene. Values represent mean ± SD of the three independent experiments. *p*-values were determined with a two-sided unpaired *t*-test. ns: not significant; * *p* ≤ 0.05; ** *p* ≤ 0.01. Effects of BOT in combination with 2DG on cell viability are provided in Appendix A. (**D**) Simplified scheme of glycolysis and pentose phosphate pathway. The targets for 2DG and BOT are depicted in red. The scheme was created with BioRender.com (accessed on 18 August 2021).

## Data Availability

The data presented in this study are available in article and Appendix A.

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
