# Peer review of "Targeting the Pentose Phosphate Pathway for SARS-CoV-2 Therapy"

_metabolites, 2021, doi:10.3390/metabo11100699_

Round 1

Reviewer 1 Report

In the manuscript entitled, “Targeting the pentose phosphate pathway for SARS-CoV-2 therapy”, Bojkova et al identified that the non-oxidative Pentose Phosphate Pathway increased during SARS CoV-2 infection in Caco-2 cell culture models. They investigated the role of TKT inhibitor benfooxythiamine (BOT) against SARS-CoV-2 replication and that found that the application of BOT increased further the potential of glycolysis inhibitor 2DG. It is a clear study of concomitant applications of both BOT and 2DG as an improved COVID-19 therapy out comes for their antiviral and/ or immunomodulatory effects. The report is novel and has solid data, but lacks of appropriate control samples in various experiments is common. Control cell layer with and without treatment lacking infection should be included in each study. In the absence of control samples, results are not so conclusive. For example Fig 2C, 2E, authors claimed increase in rate of glycolysis and ECAR during SARS CoV-2 infection on Caco-2 cell lines. However, their 0 hrs post infection stage between infected and mock samples are showing substantial differences and such differences persist through out. The data demonstrating toxicity effects of BOT and BOT+2DG alone on cells line is not available to justify their application at given concentration. And in my opinion, the role of BOT or other PPP inhibitor need to be validation in multiple cell lines and a mouse infection model. The absence of these all controls and data from other cell monolayers are the major limitation having with this study.   

Author Response

Authors’ statement: All changes are highlighted in yellow in the manuscript.

In the manuscript entitled, “Targeting the pentose phosphate pathway for SARS-CoV-2 therapy”, Bojkova et al identified that the non-oxidative Pentose Phosphate Pathway increased during SARS CoV-2 infection in Caco-2 cell culture models. They investigated the role of TKT inhibitor benfooxythiamine (BOT) against SARS-CoV-2 replication and that found that the application of BOT increased further the potential of glycolysis inhibitor 2DG. It is a clear study of concomitant applications of both BOT and 2DG as an improved COVID-19 therapy out comes for their antiviral and/ or immunomodulatory effects.

The report is novel and has solid data, but lacks of appropriate control samples in various experiments is common. Control cell layer with and without treatment lacking infection should be included in each study. In the absence of control samples, results are not so conclusive. For example Fig 2C, 2E, authors claimed increase in rate of glycolysis and ECAR during SARS CoV-2 infection on Caco-2 cell lines. However, their 0 hrs post infection stage between infected and mock samples are showing substantial differences and such differences persist through out.

Authors’ response:

Non-infected controls and infected controls were used in all experiments. In Figure 2C and 2E, the measurement started after a 24h pre-treatment period, which is described in the Methods section (4.3, p. 8, lines 252-253):

“Caco-2 cells were infected with SARS-CoV-2/FFM7 or UV-treated SARS-CoV-2/FFM7 at an MOI of 0.01 for 24 h before measurement.”

Moreover, the legend of Figure 2 was amended as follows (p. 4, lines 92-93):

“Caco-2 cells were infected with untreated or UV-inactivated SARS-CoV-2/FFM7 at an MOI of 0.01 for 24 h before the measurement.”

The data demonstrating toxicity effects of BOT and BOT+2DG alone on cells line is not available to justify their application at given concentration.

Authors’ response:

Effects on cell viability are now provided in Figure S1. Figure S1 is introduced in the legend of Figure 4 (p. 6, lines 148-149):

“Effects of BOT in combination with 2DG on cell viability are provided in Figure S1.”

And in my opinion, the role of BOT or other PPP inhibitor need to be validation in multiple cell lines and a mouse infection model. The absence of these all controls and data from other cell monolayers are the major limitation having with this study.

Authors’ response:

To address this comment, we studied the impact of BOT on SARS-CoV-2 replication in air-liquid interface cultures of primary bronchial epithelial cells and found similar effects. The Results section was amended as follows (p. 5, lines 109-112):

“Moreover, BOT inhibited replication of SARS-CoV-2 FFM7 in air-liquid interface (ALI) cultures of primary bronchial epithelial cells as indicated by the quantification of viral genomic RNA copy numbers (Figure 3D-F).”

The respective findings are presented in Figure 3D-F. The legend of Figure 3 was amended as follows (p. 5, lines 120-124):

“(D) BOT treatment strategy of air-liquid interface (ALI) cultures of primary human bronchial epithelial (HBE) cells infected with SARS-CoV-2/FFM7 (MOI 1). (E) Quantification of viral genomic RNA copy numbers in apical washes 5 days post infection. Bars represent mean ± SD of three biological replicates. (F) Cytotoxicity assay measuring LDH levels in apical washes of BOT-treated HBE ALI cultures in comparison to untreated control. Bars display mean ± SD of three biological replicates.”

Moreover, the Methods section was amended as follows (4.4, p. 9, lines 267-278):

“Primary human bronchial epithelial (HBE) cells were isolated from the lung explant tissue of a patient with lung emphysema as described previously [47]. The use of tissue was approved by the ethics committee of the Hannover Medical School (MHH, Hannover, Germany, number 2923–2015) and complies with The Code of Ethics of the World Medical Association. For differentiation into air-liquid interface (ALI) cultures, the cells were thawed and passaged once in PneumaCult-Ex Medium (StemCell technologies) and then seeded on transwell inserts (12 well plate, Sarstedt) at 4x104 cells/insert. After reaching confluence, medium on the apical side of the transwell was removed and medium in the basal chamber was replaced with PneumaCult ALI Maintenance Medium (StemCell Technologies) including Antibiotic/Antimycotic solution (Sigma Aldrich) and MycoZap Plus PR (Lonza). Criteria for successful differentiation were the development of ciliary movement, an increase in transepithelial electric resistance and mucus production.”

4.5, p. 9, lines 289-298:

“HBE cells in ALI cultures were infected with SARS-CoV-2/FFM7 at an MOI of 1 from the apical site. The inoculum was removed after 2 h, and cells were washed three times with PBS. For antiviral drug testing, the compounds were added after the infection period from both the apical and the basal site. The apical treatment was removed after one day. Genomic viral RNA copy numbers were determined after five days. Cytotoxicity was determined by LDH-Glo™Cytotoxicity Assay (Promega) according to the manufacturer´s protocol. Briefly, five days post infection ALI cultures were washed apically with PBS for 30 min and 10 µl of the washing step were diluted 1/10 in LDH storage buffer and stored at -80°C until measurement. After thawing, measurement was performed using plate reader Infinite 200 (Tecan).”

Reviewer 2 Report

With interest, I read the manuscript metabolites-1372053.

Comments (no special order):

1. Lines 43-45. Please, explain. Are the changes evoked by the virus supposed to favor its replication? And if so, would blocking/reverting those changes stop/reduce virus replication? Is it the therapeutic mechanism you suggest? Please, make the hypothesis clear already here.

2. All abbreviations used in the Figures, not including protein names, should be explained in the legends.

3. What are the limitations of the study?

4. What would be further steps? Animal experiments? Organoids?

5. In the Discussion, please, mention other, maybe even more direct, options of antiviral treatment, such as antisense approaches developed for other types of respiratory viruses including HRV or RSV (PMID: 27155029 and 30114391).

6. Lines 175-178. Can you imagine any escpae mutation making it possible for the virus avoid the consequences of your treatment strategy.

Author Response

Authors’ statement: All changes are highlighted in yellow in the manuscript.

With interest, I read the manuscript metabolites-1372053.

Comments (no special order):

  1. Lines 43-45. Please, explain. Are the changes evoked by the virus supposed to favor its replication? And if so, would blocking/reverting those changes stop/reduce virus replication? Is it the therapeutic mechanism you suggest? Please, make the hypothesis clear already here.

Authors’ response:

To address this comment, the Introduction was amended as follows (p. 1/2, lines 46-51):

“At the cellular level, SARS-CoV-2 induces a shift of the host cell metabolism towards glycolysis in infected cells. The glycolysis inhibitor 2-deoxy-D-glucose (2DG), which targets hexokinase (the rate-limiting enzyme in glycolysis), interferes with SARS-CoV-2 infection in colon adenocarcinoma (Caco-2) cells and monocytes [8,9], suggesting that the observed metabolic changes support virus replication and represent an antiviral drug target.”

  1. All abbreviations used in the Figures, not including protein names, should be explained in the legends.

Authors’ response:

This was done.

  1. What are the limitations of the study?
  2. What would be further steps? Animal experiments? Organoids?

Authors’ response:

We feel that these are two closely related comments that can be answered together. In the course of the revision, we have added data showing that benfooxythiamine (BOT) also inhibits SARS-CoV-2 replication in air-liquid interface (ALI) cultures of primary human bronchial epithelial (HBE) cells (new Figure 3D-F). Hence, animal experiments would be a logical next step and we amended the Discussion as follows (p. 8, lines 214-215):

“Therefore, the further testing of such therapy strategies in animal models is warranted.”

  1. In the Discussion, please, mention other, maybe even more direct, options of antiviral treatment, such as antisense approaches developed for other types of respiratory viruses including HRV or RSV (PMID: 27155029 and 30114391).

Authors’ response:

To address this comment, we amended the Discussion as follows (p. 7, lines 207-203):

“Moreover, other direct antiviral approaches such as antisense strategies are under de-velopment for respiratory viruses such as SARS-CoV-2 [39-41], but are not yet clinically available as treatment option for COVID-19.”

  1. Lines 175-178. Can you imagine any escpae mutation making it possible for the virus avoid the consequences of your treatment strategy.

To address this comment, we amended the Discussion as follows (p. 7, lines 205-207):

“In this context strategies that target host cell instead of viral targets have been postulated to be potentially associated with reduced resistance formation [42,43].”

Round 2

Reviewer 1 Report

The revised manuscript is substantially improvised and acceptable form.